# Experience with Rifabutin-Containing Therapy in 500 Patients from the European Registry on *Helicobacter pylori* Management (Hp-EuReg)

**DOI:** 10.3390/jcm11061658

**Published:** 2022-03-16

**Authors:** Olga P. Nyssen, Dino Vaira, Ilaria Maria Saracino, Giulia Fiorini, María Caldas, Luis Bujanda, Rinaldo Pellicano, Alma Keco-Huerga, Manuel Pabón-Carrasco, Elida Oblitas Susanibar, Alfredo Di Leo, Giuseppe Losurdo, Ángeles Pérez-Aísa, Antonio Gasbarrini, Doron Boltin, Sinead Smith, Perminder Phull, Theodore Rokkas, Dominique Lamarque, Anna Cano-Català, Ignasi Puig, Francis Mégraud, Colm O’Morain, Javier P. Gisbert

**Affiliations:** 1Centro de Investigación Biomédica en Red de Enfermedades Hepáticas y Digestivas (CIBERehd), Hospital Universitario de La Princesa, Instituto de Investigación Sanitaria Princesa (IIS-IP), Universidad Autónoma de Madrid (UAM), 28006 Madrid, Spain; opn.aegredcap@aegastro.es (O.P.N.); m.caldas.a@gmail.com (M.C.); 2Department of Surgical and Medical Sciences, IRCCS S. Orsola, University of Bologna, 40138 Bologna, Italy; berardino.vaira@unibo.it (D.V.); saracinoilariamaria@gmail.com (I.M.S.); giulia.fiorini@aosp.bo.it (G.F.); 3Hospital Donostia, Instituto Biodonostia, Centro de Investigación Biomédica en Red de Enfermedades Hepáticas y Digestivas (CIBERehd), Universidad del País Vasco (UPV/EHU), 20014 San Sebastián, Spain; medik@telefonica.net; 4Unit of Gastroenterology, Molinette Hospital, 10126 Turin, Italy; rpellicano@cittadellasalute.to.it; 5Servicio de Gastroenterolgía, Hospital de Valme, 41014 Sevilla, Spain; almakh94@hotmail.com (A.K.-H.); mpabon@cruzroja.es (M.P.-C.); 6Unit of Gastroenterology, Consorci Sanitari de Terrassa, 08221 Terrassa, Spain; eosusanibar@gmail.com; 7Section of Gastroenterology, Department of Emergency and Organ Transplantation, University Hospital Policlinico Consorziale, 70124 Bari, Italy; alfredo.dileo@uniba.it (A.D.L.); giuseppelos@alice.it (G.L.); 8Agencia Sanitaria Costa del Sol, Red de Investigación en Servicios de Salud en Enfermedades Crónicas (REDISSEC), 29651 Marbella, Spain; drapereza@hotmail.com; 9Medicina Interna, Fondazione Policlinico Universitario A, Gemelli IRCCS, Università Cattolica del Sacro Cuore, 00168 Roma, Italy; antonio.gasbarrini@unicatt.it; 10Division of Gastroenterology, Rabin Medical Center, Sackler School of Medicine, Tel Aviv University, Tel Aviv 49100, Israel; dboltin@gmail.com; 11Faculty of Health Sciences, Trinity College Dublin, D02PN40 Dublin, Ireland; smithsi@tcd.ie (S.S.); colmomorain@gmail.com (C.O.); 12Department of Digestive Disorders, Aberdeen Royal Infirmary, Foresterhill Health Campus, Aberdeen AB25 2ZN, UK; p.s.phull@abdn.ac.uk; 13Gastroenterology Clinic, Henry Dunant Hospital, 11526 Athens, Greece; sakkor@otenet.gr; 14Hôpital Ambroise Paré, Université de Versailles St-Quentin en Yvelines, Boulogne Billancourt, 92100 Paris, France; lamarquedominique@gmail.com; 15Gastroenterology Service, Althaia Xarxa Assistencial Universitària de Manresa, 08243 Manresa, Spain; acano@aegastro.es (A.C.-C.); ignasi.puig@aegastro.es (I.P.); 16Medicine Department, Universitat de Vic-Universitat Central de Catalunya (UVicUCC), 08500 Manresa, Spain; 17INSERM U1312, Université de Bordeaux, 33076 Bordeaux, France; francis.megraud@u-bordeaux.fr

**Keywords:** *Helicobacter pylori*, *H. pylori*, rifabutin, treatment, eradication failure, culture, bismuth, rescue, Hp-EuReg

## Abstract

Background: First-line *Helicobacter pylori* (*H. pylori*) treatments have been relatively well evaluated; however, it remains necessary to identify the most effective rescue treatments. Our aim was to assess the effectiveness and safety of *H. pylori* regimens containing rifabutin. METHODS: International multicentre prospective non-interventional European Registry on *H. pylori* Management (Hp-EuReg). Patients treated with rifabutin were registered in AEG-REDCap e-CRF from 2013 to 2021. Modified intention-to-treat and per-protocol analyses were performed. Data were subject to quality control. Results: Overall, 500 patients included in the Hp-EuReg were treated with rifabutin (mean age 52 years, 72% female, 63% with dyspepsia, 4% with peptic ulcer). Culture was performed in 63% of cases: dual resistance (to both clarithromycin and metronidazole) was reported in 46% of the cases, and triple resistance (to clarithromycin, metronidazole, and levofloxacin) in 39%. In 87% of cases rifabutin was utilised as part of a triple therapy together with amoxicillin and a proton-pump-inhibitor, and in an additional 6% of the patients, bismuth was added to this triple regimen. Rifabutin was mainly used in second-line (32%), third-line (25%), and fourth-line (27%) regimens, achieving overall 78%, 80% and 66% effectiveness by modified intention-to-treat, respectively. Compliance with treatment was 89%. At least one adverse event was registered in 26% of the patients (most frequently nausea), and one serious adverse event (0.2%) was reported in one patient with leukopenia and thrombocytopenia with fever requiring hospitalisation. Conclusion: Rifabutin-containing therapy represents an effective and safe strategy after one or even several failures of *H. pylori* eradication treatment.

## 1. Introduction

*Helicobacter pylori* (*H. pylori*) is a worldwide spread bacterium that causes mainly gastritis, as well as peptic ulcer disease and gastric cancer [1]. Currently, the most used first-line therapies fail in more than 20% of cases [2]. One of the major factors affecting our ability to cure *H. pylori* infection is antibiotic resistance, mainly to the macrolide clarithromycin, which is growing dramatically in many geographic areas [3,4].

A rescue regimen including a quadruple combination of a proton pump inhibitor (PPI), bismuth, tetracycline, and metronidazole has been introduced as the optimal rescue therapy after experiencing *H. pylori* eradication failure [5]. However, this treatment results in eradication failure in at least 20% of cases [2,6,7,8]. In addition, administration of this regimen is relatively complex, is associated with a high incidence of adverse events (AEs), and many countries are currently experiencing a general unavailability of tetracycline and/or bismuth. Furthermore, mainly levofloxacin-containing rescue regimens, produce a mean eradication rate of only approximately 80%, probably due to the rising *H. pylori* resistance to quinolones [9].

And thus, even after two or more eradication treatments, *H. pylori* infection persists in several cases, and these patients constitute a therapeutic dilemma. Currently, the international guidelines recommend performing culture in the aforementioned patients to select a rescue treatment according to microbial sensitivity to antibiotics, as a standard third/fourth-line therapy is lacking, although this approach is not always practical [10]. Therefore, it seems worthwhile to perform an evaluation of drugs without cross-resistance to macrolides, nitroimidazole or quinolones as components of retreatment combination therapies [11].

Rifabutin, also known as ansamycin (LM 427), is a well-established antimicrobial agent that belongs to the S-rifamycin derivative group and has been previously successfully utilised, among others, for the treatment of atypical *Mycobacterium* infections such as *Mycobacterium avium-intracellulare* complex [12]. Rifabutin may be useful against *H. pylori* because this pathogen has high in vitro sensitivity to this drug, which does not share resistance mechanisms with clarithromycin, metronidazole, or levofloxacin [11,13,14]. Furthermore, the selection of resistant *H. pylori* strains has been low in experimental conditions [15]. Consequently, rifabutin-based rescue therapies could represent a potential and attractive strategy for *H. pylori* eradication failures [12].

The “European Registry on *H. pylori* Management” (Hp-EuReg) is an international multicentre prospective non-interventional registry starting in 2013 aimed to evaluate the decisions and outcomes in *H. pylori* management, in real-world clinical practice, by European gastroenterologists from more than 30 countries [2,16]. Therefore, the Hp-EuReg represents a valued long-lasting overview of current *H. pylori* management enabling continuous evaluation of treatment for enhancement. The present study is a sub-analysis of this large-scale international multicentre registry, which aimed to assess the effectiveness and safety of rifabutin-containing regimens used in the management of *H. pylori* in Europe.

## 2. Methods

### 2.1. European Registry on H. pylori Management (Hp-EuReg)

Hp-EuReg is an international, multicentre, prospective, non-interventional registry that has been recording information on the management of *H. pylori* infection since 2013. Hp-EuReg has a Scientific Committee that ensures coherence, continued quality and scientific integrity of the analyses performed and manuscripts produced. Additionally, the Hp-EuReg protocol [16] selected national coordinators in the 30 participating countries, where gastroenterologists are currently recruited at approximately 300 centres to provide their contribution to the study. The investigators assemble and upload a series of variables and outcomes into the registry’s database (REDCap) using an Electronic Case Report Form (e-CRF). The variables include: the patient’s demographic information; any previous attempts for eradication and the treatments employed; the outcomes of any treatment, recording details such as compliance, cure rate, follow-up, and any reported AE. The REDCap database [17] is managed and hosted by the “Asociación Española de Gastroenterología” (AEG, Madrid, Spain, www.aegastro.es, last accessed 10 March 2022), a non-profit Scientific and Medical Society that focuses on Gastroenterology research. The study was conducted according to the guidelines of the 1975 Declaration of Helsinki and was approved in 2012 by the Ethics Committee of La Princesa University Hospital (Madrid, Spain), that acted as reference Institutional Review Board, was classified by the Spanish Drug and Health Product Agency, and was prospectively registered at ClinicalTrials.gov (NCT02328131).

### 2.2. Data Analysis

Data from June 2013 to November 2021 were extracted, and a quality review was performed on all the records included for each country and centre. The PPI dose used for *H. pylori* eradication treatment was grouped into three categories as reported by Graham [18] and Kirchheiner [19]: Low dose, if the potency of PPI was between 4.5 and 27 mg omeprazole equivalents when given twice daily; standard dose, for PPI between 32 and 40 mg omeprazole equivalents when given twice daily; and high dose, for PPI between 54 and 128 mg omeprazole equivalents when given twice daily. Treatment duration was evaluated according to three categories: 10, 12 and 14 days, based on the most frequent treatment durations.

### 2.3. Effectiveness Analysis

The main outcome used to evaluate the effectiveness was the eradication rate achieved with the treatment. *H. pylori* eradication was confirmed at least one month after completing eradication treatment with at least one of the following diagnostic methods: Urea breath test, stool antigen test and/or histology.

Effectiveness was analysed in three sub-groups of patients: (1) an intention-to-treat (ITT) group that included all patients registered up to November 2021 who had at least a six-month follow-up, in this group lost to follow-up cases were deemed treatment failures; (2) a per-protocol (PP) group which included all cases that had a complete follow-up and had achieved at least 90% treatment compliance, as defined in the protocol; and (3) a modified ITT (mITT) group that aimed to reflect the closest result to that obtained in clinical practice; this group included all cases that had completed the follow-up (i.e., that had undertaken a confirmatory test after the eradication treatment), regardless of compliance.

All different treatments prescribed with rifabutin were examined *a priori* for effectiveness according to the rifabutin dose, the PPI dose, the duration of therapy and, whenever possible, to the line of treatment.

### 2.4. Statistical Analysis

Continuous variables were expressed as the mean and standard deviation, whereas qualitative variables were presented as the absolute and relative frequencies, displayed as percentages (%) and corresponding 95% confidence intervals.

A multivariate analysis (using a backward modelling strategy, and comparing models using the log-likelihood ratio) was performed to study in the mITT population the relation between the eradication rate of rifabutin-containing regimens and the following variables: age, sex (female [ref.] vs. male), indication (dyspepsia and others [ref] vs. ulcer disease), compliance (no [ref] vs. yes, defined as taking >90% of the total drug prescribed), PPI dose (low [ref.] vs. standard, and low vs. high); treatment length (10 [ref.], 12, and 14 days); treatment line (first-line [ref] vs. second-line vs. all remaining rescue therapies).

## 3. Results

### 3.1. Baseline Characteristics

Until November 2021, 500 patients from seven countries had been treated with a rifabutin-containing regimen and were registered in Hp-EuReg. Three countries accounted for the majority of cases (90% of the data): Italy (333 patients) and Spain (117 patients) followed by Israel (33 patients). The remaining participating countries registered less than 10 patients each: France (7 cases), United Kingdom (5 cases), Ireland (3 cases) and Greece (2 cases).

Mean age was 52 years (±13), 72% of them were female, 63% suffered from dyspepsia, and 4% from peptic ulcer.

The ^13^C-urea breath test represented the most frequently (86%) used non-invasive diagnostic method, while culture and antibiogram was performed in 63% of the patients. Resistance to at least one of the following antibiotics: clarithromycin, metronidazole and levofloxacin was reported in 52%, 49% and 47% of the patients, respectively. Dual resistance (to both clarithromycin and metronidazole) was reported in 46% of the cases, and triple resistance (to clarithromycin, metronidazole, and levofloxacin) in 39%. No resistance was reported in 2.2% of those patients with culture testing.

Confirmation of the eradication was at least performed by means of one of the following methods: ^13^C-urea breath test (81%), monoclonal stool antigen test (3%), histology (1.4%), polyclonal stool antigen test (0.6%), rapid urease test (0.6%), and culture (0.6%).

### 3.2. Prescriptions

In total, 18 different rifabutin-containing treatments including two or three other antibiotics in the scheme were registered. However, in 87% of the cases, rifabutin was used as part of a triple therapy together with amoxicillin and a PPI, and in an additional 6% of the patients, bismuth was added to this triple regimen, while the others correspond to different drug combinations (Table 1).

Rifabutin was mainly used in second-line (32%), third-line (25%), and fourth-line (27%) regimens; in addition, rifabutin was also used to a lesser extent as part of a first-line (9%), fifth-line (6%) or sixth-line (2%) therapy (Table 2).

Overall, the antibiotic treatments were mostly combined with low (46%) or high-dose PPIs (46%), and administered most frequently for 12 days (58%), and in lower proportion for 10 (25%) or 14 days (17.5%).

Rifabutin was prescribed at two main dosages: 150 mg once a day (56%) or 150 mg every 12 h, i.e., 300 mg daily (41%).

Triple therapy together with a PPI, amoxicillin and rifabutin was mainly prescribed in second- (35%), third- (25%) or fourth-line (24%); and in most of the cases (66%) for 12 days and with low-dose PPIs (51%).

Finally, quadruple therapy with a PPI, amoxicillin, rifabutin and bismuth was prescribed mainly in fourth-line treatment (66%), in 10-day regimens (84%) and with high-dose PPIs (78%).

### 3.3. Effectiveness

Overall mITT effectiveness in first-, second-, third, fourth-, fifth- and six-line regimens was 73% (*n* = 30/41), 78% (*n* = 108/139), 80% (*n* = 80/100), 66% (*n* = 75/114), 58% (*n* = 14/24) and 75% (*n* = 6/8), respectively, as further detailed in Table 2.

Although the overall eradication with 12 days therapy was numerically higher (78%: *n* = 188/241) than when prescribed for 10 or 14 days (69% in both cases: *n* = 79/115 and *n* = 42/61, respectively), these differences were not statistically significant (*p* = 0.78).

Similarly, the overall effectiveness of rifabutin regimens was higher when high-dose PPIs were used (85%, 152/179) as compared to low- or standard-dose PPIs (66%, 138/208 and 58%, 22/38, respectively), however these differences were not statistically significant (*p* = 0.86).

Overall effectiveness with a daily rifabutin dosage of 150 mg was higher (78%, *n* = 187/239) when compared to the 300 mg daily dosage (67%, *n* = 110/166), although the difference was not statistically significant (*p* = 0.53). On the other hand, when only naïve patients were assessed, the eradication rate was higher (100%, *n* = 4/4) in the 300 mg group than in the 150 mg one (69%, *n* = 25/36), although such difference did not reach a statistical significance (*p* = 0.73).

Figure 1 shows data on the mITT effectiveness of the two most frequent prescriptions of rifabutin-containing regimens (triple therapy with a PPI, amoxicillin and rifabutin and quadruple therapy with a PPI, amoxicillin, rifabutin and bismuth) according to treatment length, PPI dose and treatment line (from second- to fourth-line). These data are further detailed in Appendix A and compared below independently for treatment length and PPI dose.

The overall eradication rate achieved with the triple therapy with amoxicillin and rifabutin was 73% (*n* = 265/363); however, in second- and third-line (77%, *n* = 103/133 and 79%, *n* = 66/84, respectively) cure rate was higher than in fourth-line (64%, *n* = 53/83), with no statistical differences in the eradication rate between treatment lines (*p* = 0.51). Effectiveness was numerically higher (78%, *n* = 187/240) for 12-day treatment than that of 10-day (70%, *n* = 53/76) or 14-day (54%, *n* = 21/39) treatment, but these differences were not statistically significant (*p* = 0.30). Additionally, higher PPI doses provided better outcomes with this same regimen (87.5%, *n* = 119/136) than when combined with either low (66%, *n* = 129/196) or standard PPI doses (53%, *n* = 16/30), with statistically significant differences in the eradication rate (*p* = 0.000). Further details are presented in Appendix A.

The overall eradication rate achieved with the quadruple therapy with amoxicillin, rifabutin and bismuth in fourth-line was 68% (*n* = 21/31), and was similar when prescribed for a duration of 10 days (68%, *n* = 18/27) or when combined with high-dose PPIs (68%, *n* = 16/24). Likewise, no statistically significant difference (*p* = 0.20) was reported in the eradication rate with quadruple therapy with amoxicillin, rifabutin and bismuth according to treatment duration, nor PPI dose. Further details are presented in Appendix A.

### 3.4. Safety and Compliance

At least one AE was registered in 26% (*n* = 126/480) of the patients: the most common AEs were nausea (7.6%, *n* = 38)—lasting in over 80% of cases between 4 and 5 days—, and asthenia (6.2%, *n* = 31)—lasting in 79% of cases between 5 and 7 days. Intensity of both AEs was mild or moderate in all patients. One serious AE (0.2%) was reported in one patient (treated with the triple therapy with amoxicillin and rifabutin), with leukopenia, thrombocytopenia and accompanying fever that required hospitalisation; but this patient recovered spontaneously and was discharged from hospital without further complications.

Triple therapy with amoxicillin and rifabutin exhibited significantly (*p* = 0.001) lower overall AE incidence (23%, *n* = 97/417) than the quadruple rifabutin regimen with bismuth (52%, *n* = 16/31); however, the latter provided better results in terms of treatment compliance (97%, *n* = 30/31) than the triple therapy (86%, *n* = 369/416), with no significant differences (*p* = 0.13).

Overall compliance with treatment was 89% (*n* = 428), although 25% (*n* = 13, where nine were treated with the triple therapy with amoxicillin and rifabutin, and one with the quadruple with rifabutin and bismuth) of patients interrupted the treatment due to AEs.

### 3.5. Univariate Analysis

In order to equate conditions, the effectiveness of triple therapy with amoxicillin and rifabutin versus quadruple therapy with amoxicillin, rifabutin and bismuth was compared when both were prescribed for 10 days and in fourth-line treatment (that is, in those subgroups where more data were available). In this context, no statistically significant difference in mITT eradication rate was found between triple therapy and the bismuth-rifabutin quadruple regimen (66% [*n* = 27/41] vs. 67% [*n* = 14/21], *p* = 0.59).

The analysis was also stratified by PPI dose with both regimens, and a comparison could be performed when both were combined with high-dose PPIs (during 10 days and in fourth-line treatment): triple therapy reported a numerically higher mITT eradication rate (100%, *n* = 5/5) than quadruple therapy (68%, *n* = 13/19); but these differences were not statistically significant (*p* = 0.28).

Additionally, no subgroup analysis was performed by dosage schedule as a whole; however, it is noteworthy that one patient was prescribed 150 mg of rifabutin every 12 h (i.e., 300 mg rifabutin per day).

### 3.6. Multivariate Analysis

Stepwise multivariate logistic regression analysis was further performed to determine those variables influencing the most the mITT eradication rate of triple therapy with amoxicillin and rifabutin (as it was the only regimen with a sufficient sample size for multivariate analysis).

The analysis revealed that among all the covariates being analysed (age, sex, indication, compliance, PPI dose; treatment length; treatment line), compliance (OR = 13.1, 95%CI = 2.1–81.4), a high-dose PPI (OR = 4.4, 95%CI = 2.3–8.3), as well as the age (OR = 1.03, 95%CI = 1.01–1.05) were significantly associated with higher therapy success. Additionally, 14-day treatment duration was associated with a tendency (although not significant) towards lower effectiveness, as compared to 10-day therapy.

## 4. Discussion

A relevant proportion of patients still fail to eradicate *H. pylori* infection, even with the current most effective treatment regimens. Nowadays, clinicians have to take into account treatment failures apart from substantiating first-line eradication regimens. In this context, rifabutin presents potential utility against *H. pylori*.

Hp-EuReg, an international multicentre prospective non-interventional European Registry on *H. pylori* management [16], has allowed us to recruit the largest series of patients treated with rifabutin, with 500 patients included from 2013 to 2021. Our results, with approximately 80% *H. pylori* cure rate—in agreement with previous reviews [12,20,21]—could be considered relatively encouraging, especially taking into account that rifabutin regimens were prescribed mainly after two-to-four eradication failures with key antibiotics such as clarithromycin, metronidazole, tetracycline and levofloxacin. In fact, resistance to clarithromycin, metronidazole and levofloxacin was reported in 52%, 49% and 47% of our patients, respectively. Furthermore, dual resistance (to both clarithromycin and metronidazole) was reported in almost half of the cases, and triple resistance (to clarithromycin, metronidazole, and levofloxacin) in more than a third.

The encouraging results obtained with rifabutin-containing regimens are probably due to the low *H. pylori* resistance rate to this antibiotic [13,22,23]. Thus, in a previous systematic review including 39 studies and almost 10,000 patients, rifabutin resistance was reported in only 0.13% of the cases; furthermore, when only naïve *H. pylori* participants were considered, this rate was even lower (0.07%) [12]. In our study, no prevalence data on the rifabutin resistance were available, and therefore the reduced effect of the resistance to this antibiotic on the high effectiveness of treatments could not be confirmed. Resistance to rifabutin is due to mutations in the *rpo*B gene and there is no cross-reaction with the resistance mechanisms to the other antibiotics; accordingly, in cases with *H. pylori* infection with primary resistance to clarithromycin or metronidazole (or both), rifabutin therapy has been reported highly effective [24,25,26,27,28,29,30,31], and even in patients with triple resistance to these two antibiotics plus quinolones [14], which is the usual real-life scenario after several treatment eradication attempts.

When prescribed as a second-line regimen, a rifabutin-containing regimen was effective in 78% of the patients in our study; even when used as a sixth-line therapy this regimen was able to cure the infection in 75% of our cases. These results are consistent with those summarised in a previous relevant review [12]. Accordingly, a recent study found that the efficacy of rifabutin treatment was not significantly influenced by the number of previous treatment failures: eradication rates in patients with one, two, three, and four or more previous failures were 78.3%, 89.6%, 68.6%, and 88.9%, respectively (non-statistically significant differences) [32].

Regarding the specific regimen prescribed in Hp-EuReg, rifabutin was used as part of a triple therapy together with amoxicillin and a PPI in most (≈90%) of the cases, which achieved cure rates of 77% in second-line, 79% in third-line, and 64% in fourth-line treatments. Liu et al. conducted a meta-analysis of clinical trials for eradication of *H. pylori* that included a treatment arm with a PPI, rifabutin, and amoxicillin; twenty-one studies were included, and the overall reported eradication rate was 70% [20]. More recently, Gingold-Belfer et al. performed another meta-analysis of randomised controlled trials with a treatment arm consisting of PPI, amoxicillin, and rifabutin, and the pooled cure rate in the 33 studies selected was 71.8% [21].

With respect to rifabutin dose and frequency in *H. pylori* eradication regimens, the majority of studies prescribe rifabutin 300 mg/day [12]. However, in our study, approximately 50% of the patients received only 150 mg/day. The *H. pylori* cure rate in published small number of previous studies using rifabutin 150 mg/day was about 40–70% only [33,34], although other studies have reported higher eradication rates [14,29,35]. There is only one study directly comparing both doses: Perri et al. [33] performed a randomised study where patients persistently infected after one or more courses of standard regimens were treated for 10 days with pantoprazole, amoxicillin, and rifabutin 150 mg once daily or 300 mg once daily. In intention-to-treat analysis, eradication rates were 67% in the rifabutin 150 mg group and significantly higher (87%) in the rifabutin 300 mg group. As opposed to these results, our study showed a similar effectiveness with a daily rifabutin dosage of 150 mg (78%) to that of the 300 mg daily dosage (67%); however, when only naïve patients were assessed, the eradication rate was higher (100%) in the 300 mg group than in the 150 mg one (69%), although such difference remained not statistically significant. This may probably due to the small sample size (*n*= 4 patients) in our 300 mg group, which may not be very representative and therefore the lack of statistical significance may not imply the real effect.

The ideal length of rifabutin treatment remains unclear, but 10 to 12-day regimens are generally recommended [12], as it was the case in Hp-EuReg, where ≈80% of the patients received this regimen. In some reports, a 7-day course was as efficacious as 10 to 14-day regimens, while this shorter duration dramatically reduced the efficacy, with eradication rates of only 44%, as reported elsewhere [36]. Although rifabutin treatment could be more likely to be successful when treatment duration is 14 days, as suggested in some studies [21], many other exposed that therapy between 12 and 14 days achieves similar results to the 10-day course and is likely to rise the incidence of AEs [37]. A recent randomised controlled trial compared 10-day vs. 14-day eradication therapy with PPI, amoxicillin and rifabutin and determined that over 90% of patients resulted in successful eradication with 14-day therapy, but stated that considering the tolerability of therapy, 10-day treatment may be enough to obtain a successful eradication rate [38]. In line with these results, in our study, therapy with amoxicillin and rifabutin was prescribed mostly for 12 days, achieving higher effectiveness (78%) than when prescribed for 10 (70%) or even 14 days (54%), although these differences were not statistically significant.

Bismuth is one of the few antimicrobials to which resistance is not developed [39]. In addition, bismuth has an additive effect with antibiotics, overcomes levofloxacin and clarithromycin resistance and its efficacy is not affected by metronidazole resistance [39,40]. Therefore, combining bismuth and rifabutin in the same regimen may be a promising option. Some authors have evaluated a combination of a triple therapy with a PPI, amoxicillin and rifabutin, with bismuth—and thus converting this triple regimen into a quadruple one—, with encouraging results [41,42,43]. Ciccaglione et al. reported, in a small sample size population, that the addition of bismuth to a triple therapy that included PPI, amoxicillin, and rifabutin in patients treated for the third time for *H. pylori* infection resulted in 30% therapeutic gain compared to rifabutin-based triple therapy alone [42]. In our study, the addition of bismuth did not seem to increase the effectiveness of the rifabutin-triple regimen 68% vs. 73%, respectively), although the number of patients receiving the quadruple regimen was considerably low (*n* = 32) and in most of them was utilised as a fourth-line rescue regimen. And so, as previously stated this lack of statistical differences between the triple and the bismuth quadruple therapies, may be due to the differences in sample sizes between groups or potentially that the finding could reliably exclude several covariates that may introduce some bias.

In Hp-EuReg, compliance with treatment was quite satisfactory (89%) and the safety profile was acceptable. At least one AE was registered in 28% of the patients (most frequently nausea and asthenia), but most of them (>95%) were of mild-to-moderate intensity. This seems to be a favourable safety profile, mainly when compared to other well stablished eradication regimens, such as the bismuth and non-bismuth quadruple therapies [44,45]. In a recent meta-analysis of all studies including rifabutin for *H. pylori* eradication, mean rate of AEs was 15% [12]. In our study, only one serious AE (0.2%) was reported in a single patient with leukopenia and thrombocytopenia with fever requiring hospitalisation. Accordingly, a recent review on the use of rifabutin for *H. pylori* infection [21], only found one severe AE reported in the literature [31].

Myelotoxicity is the most significant AE of rifabutin [46,47]. Overall, this complication is rare and is far more likely when high dose (600 mg/day) and prolonged duration therapy is used [46,47]. Several cases of myelotoxicity during *H. pylori* therapy have been mentioned in the literature [27,28,35,37,48,49,50,51,52]. However, myelotoxicity was not reported in most of the studies evaluating rifabutin for *H*. *pylori* infection. In the studies describing this complication, myelotoxicity was observed in 1.5% to 3% of the patients [27,28,37,50,52], although some studies reported higher incidence [49,51]. In the meta-analysis conducted by Gingold-Belfer et al., neutropenia was addressed in 27 studies, where 19 of them reported no cases of neutropenia and eight studies reported at least one case; and so, in total, only 17 patients developed neutropenia across all studies [21]. All patients reported in the literature recovered from myelotoxicity uneventfully within few days, with spontaneous recovery from leukopenia. In several cases, the leukopenia was clinically apparent with fever [37,49]. Infections or other adverse outcomes related to reduced white cell count have not been reported in the setting of *H. pylori* treatment [27,28,37,48,49,50,51,52].

Only few studies have compared a triple combination of a PPI, amoxicillin and rifabutin with the widely used “classic” bismuth quadruple regimen [27,30,33,36]. As an example, in the randomised study by Perri et al., side effects were less frequent in rifabutin-treated patients than in those on bismuth quadruple therapy; additionally, the eradication rates were reported the same in both groups (67%) when rifabutin was prescribed at 150 mg, and even higher (87%) when prescribed at 300 mg [33]. Miehlke et al. [30] compared, also in a randomised study, rifabutin-based triple therapy for 7 days vs. high-dose dual therapy for 14 days, for rescue treatment of *H. pylori*; premature discontinuation of treatment occurred in 2% and 5% of patients respectively. Finally, one study directly compared rifabutin to levofloxacin as third-line therapy for *H. pylori*, and AEs were reported in 60% and 50% of the cases, respectively. However, in this study, the intention-to-treat eradication rate was reported lower (45%) in the rifabutin group as compared to the levofloxacin one (85%) [51].

The major limitation of our study was that the rifabutin regimens in the studied cohort were heterogeneous, including several schemes, doses, and durations. Nonetheless, most of the regimens included a triple combination of rifabutin together with a PPI and amoxicillin administered for 10-to-12 days. Heterogeneity was inherent to the study design of Hp-EuReg (i.e., observational, non-interventional) and therefore difficult to avoid, as wide selection criteria were initially established to reflect as much as possible real clinical practice. Most of patients came from only three countries, and this might introduce selection bias. Another point to highlight is that culture was performed in only 63% of the cases; however, this reflects real routine gastroenterology practice in Europe, where antibiograms are not performed on a routine basis and treatments are mainly empirically prescribed [10]; furthermore, as previously noted, resistance to rifabutin is exceptional, occurring in less than 1% of the cases [12].

In summary, from the analysis of Hp-EuReg, it can be deduced that rifabutin-containing therapy represents a relatively effective and safe strategy after one or even several failures of *H. pylori* eradication treatment, although still insufficient as not reaching the desired 90% threshold of eradication [7]. Since resistance to rifabutin is practically non-existent, and rifabutin therapy is highly effective even in patients with primary resistance to clarithromycin, metronidazole, and levofloxacin, rifabutin usage in an empirical manner may be suggested as “rescue” therapy without culture in those patients in whom these antibiotics have failed. Recent studies have also evaluated the role of rifabutin in *H. pylori* treatment in naïve patients, with encouraging results [31,53]. Nevertheless, the consideration of rifabutin as a novel first-line treatment option for *H. pylori* infection should be carefully weighed against concerns regarding microbial resistance, treatment cost (rifabutin is quite expensive), and the availability and effectiveness of alternative drugs.

## Figures and Tables

**Figure 1 jcm-11-01658-f001:**
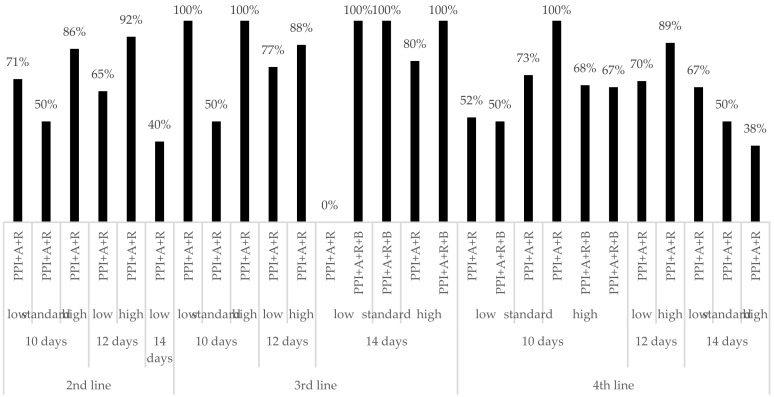
Effectiveness by modified intention-to-treat analysis of both triple therapy with amoxicillin and rifabutin and quadruple therapy with amoxicillin, rifabutin and bismuth, according to duration, potency of acid inhibition and treatment line. A: amoxicillin; B: bismuth; R: rifabutin; PPI: proton pump inhibitor (low-dose PPI: 4.5–27 mg omeprazole equivalents (OE) twice daily (bid) (i.e., 20 mg OE bid), standard-dose PPI: 32–40 mg omeprazole equivalents bid (i.e., 40 mg OE bid), high-dose PPI: 54–128 mg omeprazole equivalents bid (i.e., 60 mg OE bid).

**Table 1 jcm-11-01658-t001:** Rifabutin-containing regimens registered in Hp-EuReg between 2013 and 2021.

Prescriptions	*n* (%)
**Triple-PPI + A+R**	434 (86.8)
**Quadruple-PPI + A+R + B**	32 (6.4)
**Triple-PPI + R+Tc**	5 (1.0)
**Triple-PPI + L+R**	4 (0.8)
**Quadruple-PPI + R+D + B**	4 (0.8)
**Triple-PPI + M+R**	3 (0.6)
**Quadruple-PPI + L+R + Tc**	3 (0.6)
**Quadruple-PPI + R+B + minocycline**	3 (0.6)
**Triple-PPI + R+minocycline**	2 (0.4)
**Quadruple-PPI + A+R + Tc**	2 (0.4)
**Quadruple-PPI + A+R**	1 (0.2)
**Triple-PPI + R+D**	1 (0.2)
**Dual-PPI + R**	1 (0.2)
**Triple-PPI + C+R**	1 (0.2)
**Quadruple-PPI + L+R + B**	1 (0.2)
**Quadruple-PPI + C+R + B**	1 (0.2)
**Quadruple-PPI + A+L + R+Tc**	1 (0.2)
**Sequential-PPI + C+A + R**	1 (0.2)
**Total**	500 (100)

A: amoxicillin; B: bismuth; C: clarithromycin; D: doxycycline; L: levofloxacin; M: metronidazole; R: rifabutin; Tc: tetracycline; PPI: proton pump inhibitor; N: Number of patients with prescribed treatment.

**Table 2 jcm-11-01658-t002:** Rifabutin-containing prescriptions and overall effectiveness according to treatment line.

	Use, *n* (%)	mITT, *n* (%)	95%CI	PP, *n* (%)	95%CI
**Total**	500 (100)	426 (73.5)	69–77	415 (74)	70–78
**1st line**	43 (9)	41 (73)	58–88	41 (73)	58–88
**2nd line**	160 (32)	139 (78)	70–85	136 (78)	71–85
**3rd line**	124 (25)	100 (80)	72–88	97 (81)	73–90
**4th line**	134 (27)	114 (66)	57–75	109 (67)	58–76
**5th line**	29 (5)	24 (58)	36–79	24 (58)	36–80
**6th line**	10 (2)	8 (75)	35–97	8 (75)	35–97

mITT: modified intention-to-treat; PP: per protocol, CI: confidence interval, *n*: total number of patients analysed.

## Data Availability

The data that support the findings of this study are not publicly available given that containing information could compromise the privacy of research participants. However, previous published data on the Hp-EuReg study, or de-identified raw data referring to current study, as well as further information on the methods used to explore the data could be shared, with no particular time constraint. Individual participant data will not be shared.

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
