# Peer review of "Experience with Rifabutin-Containing Therapy in 500 Patients from the European Registry on Helicobacter pylori Management (Hp-EuReg)"

_jcm, 2022, doi:10.3390/jcm11061658_

Round 1

Reviewer 1 Report

Extensive and relevant international, non-interventionist study on the use of rifabutin in the treatment and retreatment of H. pylori infection in European countries by using Hp-Eureg. The large sample size and high level of information regarding rifabutin regimes and duration of treatment constitute the strengths of the study, although antibiotic susceptibility data were obtained from 63% of the patients.

In the results section, additional information could be addressed on:

  • The percentage of the methods employed to confirm H.  pylori eradication in the study population.
  • Given the importance of rifabutin myelotoxicity, a more detailed description of the evolution of the affected patient.

Author Response

Reviewer: 1

Extensive and relevant international, non-interventionist study on the use of rifabutin in the treatment and retreatment of H. pylori infection in European countries by using Hp-Eureg. The large sample size and high level of information regarding rifabutin regimes and duration of treatment constitute the strengths of the study, although antibiotic susceptibility data were obtained from 63% of the patients.

In the results section, additional information could be addressed on:

  • The percentage of the methods employed to confirm H.  pylori eradication in the study population.

Thank you for your comment. These data have been added under the results’ heading 3.1., at the end of the second paragraph (lines 170-173).

  • Given the importance of rifabutin myelotoxicity, a more detailed description of the evolution of the affected patient.

Thank you for the suggestion. The patient suffering from leukopenia, thrombocytopenia and fever requiring hospitalisation, recovered spontaneously and was discharged from hospital without further complications. This information has been added in manuscript (lines 252-253).

Reviewer 2 Report

Comments to the Author:

In this registry-based cohort study, the authors included 500 patients with Helicobacter pylori infection who were treated with a rifabutin-containing regimen among the participants who were registered for the Hp-EuReg. The authors showed that the rifabutin-containing regimen was mainly used as triple therapy for salvage treatment of H. pylori eradication in real-world practice in three countries of Europe. The results were of clinical importance and the methods were valid. However, several points should be addressed including outcome reporting and interpretation.

  1. Outcome reporting and interpretation.

The most important problem of this study is that the authors reported the difference between different treatment regimens or patient groups as if there were difference while there was actually not. They used the expression ‘higher’ or ‘lower’ with “but differences were not statistically significant”. These expressions are not scientifically correct and should not be used. In addition, it is recommended to add the denominators and numerators when they report percentages unless the numbers can be found in the table.

The results parts (lines 197-206) should be revised. For example, “When treatments were prescribed for 12 days, the overall eradication was higher (78%) than when prescribed for 10 or 14 days (69% in both cases), but differences were not statistically significant.”(lines 197-199) should be revised to “There was no significant difference in the overall eradication according to the treatment duration (12 days, 78%, n/N; 10 days, 69%, n/N; and 14 days, 69%, n/N; p = 0.xxx).

  1. Outcome reporting

Please provide p-values for the lines 224 – 227 and lines 245 – 248.

  1. Outcome reporting

Please revise the expression in lines 263-264, lines 338-341, and lines 354-356. For example, in the expressions in lines 263-264. “triple therapy demonstrated higher mITT eradication rate (100%, n=5) than quadruple therapy (68%, n=19); however, these differences were not statistically significant (p=0.28). As commented above, this difference can happen by chance because the numbers are very small. The expression “demonstrated higher” should be removed.

  1. Outcomes of culture and antibiogram

The authors reported the prevalence of antibiotic-resistant H. pylori strains in the Results (lines 167-169) and the Discussion (lines 292 – 294). The latter part needs to be moved to the Results section. In addition, please describe the culture outcomes more in detail.

  1. Outcomes between empirical treatment and tailored treatment.

The authors reported joint outcomes for empirical treatment and tailored treatment. Was there any difference in the treatment regimens or outcomes between empirical and tailored treatment? Please present the outcomes of empirical treatment and tailored treatment separately.

  1. Outcomes according to the resistance profile.

There were 46% of dual resistance and 39% triple resistance. Was there any difference in the treatment regimens or outcomes between dual and triple resistance?

  1. Multivariable analysis

Please provide a table (as a main table or supplement) for the univariable and multivariable analysis for the mITT eradication rate. It is recommended to present clearly which variables were included and which variables were not in the multivariable analysis.

  1. Discussion

In lines 327 – 341, the authors described the discrepancy between literature and the current study in the association between rifabutin dose and eradication rate. However, there was no explanation or speculation why the authors think this discrepancy occurred. Please add the authors’ interpretation.

  1. Discussion

In lines 357 – 369, the authors did not provide any explanations for the discrepancy between literature and the current study in the different performance between rifabutin-based triple and quadruple therapies. Please add the authors’ interpretation.

  1. Discussion

In lines 394 – 402, the authors compared the adverse events of rifabutin-based therapy against bismuth quadruple or levofloxacin triple therapy. Please add the outcomes of eradication rate of rifabutin-based therapy against bismuth quadruple or levofloxacin triple therapy.

Author Response

Reviewer: 2

In this registry-based cohort study, the authors included 500 patients with Helicobacter pylori infection who were treated with a rifabutin-containing regimen among the participants who were registered for the Hp-EuReg. The authors showed that the rifabutin-containing regimen was mainly used as triple therapy for salvage treatment of H. pylori eradication in real-world practice in three countries of Europe. The results were of clinical importance and the methods were valid. However, several points should be addressed including outcome reporting and interpretation.

  1. Outcome reporting and interpretation.

The most important problem of this study is that the authors reported the difference between different treatment regimens or patient groups as if there were difference while there was actually not. They used the expression ‘higher’ or ‘lower’ with “but differences were not statistically significant”. These expressions are not scientifically correct and should not be used. In addition, it is recommended to add the denominators and numerators when they report percentages unless the numbers can be found in the table.

The results parts (lines 197-206) should be revised. For example, “When treatments were prescribed for 12 days, the overall eradication was higher (78%) than when prescribed for 10 or 14 days (69% in both cases), but differences were not statistically significant.”(lines 197-199) should be revised to “There was no significant difference in the overall eradication according to the treatment duration (12 days, 78%, n/N; 10 days, 69%, n/N; and 14 days, 69%, n/N; p = 0.xxx).

Thank you for your critical comment. The reporting of the outcomes has been reviewed and rewritten following your suggestions, please check lines 205-279 in the updated manuscript version.

  1. Outcome reporting

Please provide p-values for the lines 224 – 227 and lines 245 – 248.

Thank you for your comment, p-values have been provided as suggested, please check lines 244-255 in the updated manuscript version.

  1. Outcome reporting

Please revise the expression in lines 263-264, lines 338-341, and lines 354-356. For example, in the expressions in lines 263-264. “triple therapy demonstrated higher mITT eradication rate (100%, n=5) than quadruple therapy (68%, n=19); however, these differences were not statistically significant (p=0.28). As commented above, this difference can happen by chance because the numbers are very small. The expression “demonstrated higher” should be removed.

Thank you, we agree, this has been amended as appropriate, please check lines 284-295 in the updated version of the manuscript.

  1. Outcomes of culture and antibiogram

The authors reported the prevalence of antibiotic-resistant H. pylori strains in the Results (lines 167-169) and the Discussion (lines 292 – 294). The latter part needs to be moved to the Results section. In addition, please describe the culture outcomes more in detail.

Thank you very much for your observation. Results reported in the discussion section have now been addressed in the results section; however, we have decided not to remove the information from the discussion as it was in line with previous argument. Additionally, further outcome data on culture testing have been added as suggested, but please be aware information of culture testing outcomes in the Hp-EuReg is scarce and limited due to two main reasons: 1) the CRD does not collect further information on the culture method used and 2) overall in the registry, there is only a 10% data reporting on culture testing with an antibiotic resistance test result available. Please check lines 175-179.

  1. Outcomes between empirical treatment and tailored treatment.

The authors reported joint outcomes for empirical treatment and tailored treatment. Was there any difference in the treatment regimens or outcomes between empirical and tailored treatment? Please present the outcomes of empirical treatment and tailored treatment separately.

Thank you very much for your comment. Actually, there was a mistake in our manuscript and we have removed the sentence you are mentioning: “The effectiveness analyses were performed jointly for patients treated empirically or when treatment was based on the testing of bacterial resistance (i.e., antibiogram, as performed in routine clinical practice in each local centre”.

In fact, in our study, no tailored treatments were prescribed as all treatments included prescribed amoxicillin and rifabutin (+/- bismuth), two antibiotics with almost no bacterial resistance; and therefore, all treatments were all empirically prescribed and were not based on the results of the bacterial antibiotic susceptibility.

  1. Outcomes according to the resistance profile.

There were 46% of dual resistance and 39% triple resistance. Was there any difference in the treatment regimens or outcomes between dual and triple resistance?

Thank you for your comment, please refer to the response above. Treatments’ effectiveness was not based on the antibiotic resistance test.

  1. Multivariable analysis

Please provide a table (as a main table or supplement) for the univariable and multivariable analysis for the mITT eradication rate. It is recommended to present clearly which variables were included and which variables were not in the multivariable analysis.

Thank you for your comment. We agree variables need to be presented clearly and following your suggestion we have rewritten the paragraph (please check lines 300-312). All the necessary information has been now included within the text of the manuscript and we believe the table is no longer necessary.

  1. Discussion

In lines 327 – 341, the authors described the discrepancy between literature and the current study in the association between rifabutin dose and eradication rate. However, there was no explanation or speculation why the authors think this discrepancy occurred. Please add the authors’ interpretation.

Thank you very much for your comment. In our study, this lack of differences between groups may be due to the small sample size in the 300 mg group (5 naïve patients only), reducing the real effect of the outcome measure. A sentence clarifying this has been added in the manuscript, please check lines 377-379.

  1. Discussion

In lines 357 – 369, the authors did not provide any explanations for the discrepancy between literature and the current study in the different performance between rifabutin-based triple and quadruple therapies. Please add the authors’ interpretation.

Thank you very much for your comment. As previously stated, we believe one of the main reasons for this difference, may be the lack of power in our statistical comparison, due to the reduced sample size in one group as compared to the other; and so, the findings may be excluding some potential other covariates responding to a type II error. A sentence clarifying this situation has been added as appropriate, please check lines 411-414.

  1. Discussion

In lines 394 – 402, the authors compared the adverse events of rifabutin-based therapy against bismuth quadruple or levofloxacin triple therapy. Please add the outcomes of eradication rate of rifabutin-based therapy against bismuth quadruple or levofloxacin triple therapy.

Thank you very much for your suggestion. The eradication rates have been added; please check both lines 442-445 and 450-452.

Reviewer 3 Report

Overall, I believe that the research presented in this article is well-conducted and clinically relevant. There are, however, a few issues that need to be improved. I kindly ask you to respond to the following issues:

Minor amendments:

  • The name Helicobacter pylori / pylori in the abstract should be written in italics
  • Line 81: Mycobacterium infections
  • Line 82: Mycobacterium avium-intracellulare complex
  • Lines 161-163: The total number of patients is not 500 (Italy + Spain + Israel). If other countries were also included, they should also be listed.
  • Lines 254-255: Is this sentence cut short or is it a title? If that was a title, it should be added directly in point 3.5.

Major amendments:

  • I believe that Table 3 should be transferred to Supplementary materials and should be replaced with 2-3 graphs that will illustrate the results briefly. There are too many results in this table, making it impossible to read them easily.
  • I believe that the narratives should be slightly modified and say that the degree of eradication of rifabutin-containing therapy is insufficient (it reaches an average of about 75%). pylori was recognized as an infectious agent several years ago, and therefore the approach to treating this pathogen has changed in recent years. It is suggested that the minimum eradication threshold required is 90%. Therapies that do not meet this condition are insufficient. Please see the publications, among others:

Clin Gastroenterol Hepatol. 2021 Mar 26;S1542-3565(21)00336-0. doi: 10.1016/j.cgh.2021.03.026.

https://pubmed.ncbi.nlm.nih.gov/33775895/

Antibiotics (Basel). 2020 Oct 3;9(10):671. doi: 10.3390/antibiotics9100671.

https://pubmed.ncbi.nlm.nih.gov/33023041/

Author Response

Reviewer: 3

Overall, I believe that the research presented in this article is well-conducted and clinically relevant. There are, however, a few issues that need to be improved. I kindly ask you to respond to the following issues:

Minor amendments:

  • The name Helicobacter pylori / pylori in the abstract should be written in italics

Thank you for the observation. In the original manuscript submitted, the format was correct and probably it has been a transfer edition mistake. This has been amended now in the last updated version of the manuscript.

  • Line 81: Mycobacterium infections.

Amended, thank you

  • Line 82: Mycobacterium avium-intracellulare complex.

Amended, thank you

  • Lines 161-163: The total number of patients is not 500 (Italy + Spain + Israel). If other countries were also included, they should also be listed.

Thank you for your comment. You are right Italy + Spain + Israel account for 90% of the data as stated in the manuscript; however, following your suggestion, the remaining participating countries have been listed for information (lines 163-164).

  • Lines 254-255: Is this sentence cut short or is it a title? If that was a title, it should be added directly in point 3.5.

Thank you for your observation. It was stated as a title; but we fully agree it appears as a cut short sentence. We have therefore preferred to embed it in the text directly and leave the title as it was (please, check amendment line 265-266). In the same context, we have removed subtitles under paragraph 3.3 for the same reasons and for consistency with the manuscript formatting (please check modifications lines 230 and 240).

Major amendments:

  • I believe that Table 3 should be transferred to Supplementary materials and should be replaced with 2-3 graphs that will illustrate the results briefly. There are too many results in this table, making it impossible to read them easily.

Thank you for your suggestion. We agree and we have therefore included a simplified figure within the manuscript and transferred the detailed table to the supplementary material.

  • I believe that the narratives should be slightly modified and say that the degree of eradication of rifabutin-containing therapy is insufficient (it reaches an average of about 75%). pylori was recognized as an infectious agent several years ago, and therefore the approach to treating this pathogen has changed in recent years. It is suggested that the minimum eradication threshold required is 90%. Therapies that do not meet this condition are insufficient. Please see the publications, among others:

Clin Gastroenterol Hepatol. 2021 Mar 26;S1542-3565(21)00336-0. doi: 10.1016/j.cgh.2021.03.026.

https://pubmed.ncbi.nlm.nih.gov/33775895/

Antibiotics (Basel). 2020 Oct 3;9(10):671. doi: 10.3390/antibiotics9100671.

https://pubmed.ncbi.nlm.nih.gov/33023041/

Thank you for your critical appraisal. We have slightly modified the text in our conclusions in line with your opinion (lines 426-428).

Round 2

Reviewer 2 Report

It seems that the initial comments were appropriately responded.
No further comments on this manuscript.